# Data Pruning of Tomographic Data for the Calibration of Strain Localization Models

**William Hilth [1], David Ryckelynck [1,\*] and Claire Menet [2]**

[1] MAT—Centre des Matériaux, Mines ParisTech, PSL Research University, CNRS UMR 7633, 10 rue Desbruères, 91003 Evry, France; william.hilth@mines-paristech.fr

[2] University of Lyon, INSA de Lyon, UMR CNRS 5510, 20 Avenue Albert Einstein, 69100 Villeurbanne, France; claire.menet@montupet-group.com

\* Correspondence: david.ryckelynck@mines-paristech.fr

**Abstract:** The development and generalization of Digital Volume Correlation (DVC) on X-ray computed tomography data highlight the issue of long-term storage. The present paper proposes a new model-free method for pruning experimental data related to DVC, while preserving the ability to identify constitutive equations (i.e., closure equations in solid mechanics) reflecting strain localizations. The size of the remaining sampled data can be user-defined, depending on the needs concerning storage space. The proposed data pruning procedure is deeply linked to hyper-reduction techniques. The DVC data of a resin-bonded sand tested in uniaxial compression is used as an illustrating example. The relevance of the pruned data was tested afterwards for model calibration. A Finite Element Model Updating (FEMU) technique coupled with an hybrid hyper-reduction method aws used to successfully calibrate a constitutive model of the resin bonded sand with the pruned data only.

**Keywords:** archive; model reduction; 3D reconstruction; inverse problem plasticity; data science

---

## 1. Introduction

With the development and the generalization of digital image correlation (DIC) (see Chu et al. [1]) or digital volume correlation (DVC) (see Bay et al. [2]) techniques on Computed Tomography (CT) data, the volume of data acquired has drastically increased. This raises new challenges, such as data storage, data mining or the development of relevant experiments-simulations dialog methods such as model validation and model calibration.

In experimental mechanics, the access to full 3D fields such as displacement or strain fields is far richer than 1D load–displacement curves. These data can drive finite element simulations for model calibration. Although extremely convincing, the increasing resolution of the full-field measurement tools, such as X-ray Computed Tomography, leads to an explosion of the volume of data to store. The long term storage of CT datasets is nowadays an issue (see Ooijen et al. [3]).

This paper proposes a numerical method for pruning 3D dataset related to DVC when it becomes necessary to free up storage capacity. Often, when new experimental results need to be saved, storage memory must be released. The pruned data contain information similar to the original data, but with less memory required. The proposed approach aims to prune experimental data while preserving the ability to identify constitutive equations (i.e., closure equations in solid mechanics) reflecting strain localizations. It is a mechanical based approach to prune DVC data. Outside a reduced experimental domain (RED), the experimental data are deleted. Original experimental data are preserved solely in the RED. We also propose a calibration procedure whose computational complexity is consistent with the pruning of the experimental data.

Compression of data is known to be a convenient approach to restore storage capacity. For instance, MP3 files are a fairly common way to reduce the size of audio files for daily use (see Pan [4]). However, a non-negligible loss of information is needed, but controlled. The MP3 compression roughly consists in filtering certain components of the non-reduced audio file that are actually non-audible for most people. In other words, the MP3 algorithm was made to prune the audio data that are not absolutely necessary. Usually, the compression rate is around 12. In the same philosophy, there can be a way to massively compress the experimental data taken from experiments with a controlled loss of information based on an algorithm that detects the pertinent information. This has been proposed in [5] by using a sensitivity analysis with respect to variations of calibration parameters. These parameters are the coefficients of a given model that should reflect the experimental observations. The result is that the pruned data are dedicated to a given model. In this paper, a model-free approach is proposed. It aims to make possible various calibrations with different models after data pruning. Here, the relevant information are local but situated in regions submitted to strain localization. The data submitted to the pruning procedure are the outputs of a Digital Volume Correlation that reconstructs the displacement field $\mathbf{u}(\mathbf{x}, t)$ from observations at time instants $(t_j)_{j=1,\ldots,N_t}$, over a spatial domain $\Omega$, where $\mathbf{x}$ is a position vector. The geometry of the experimental sample is approximated by a mesh and the determined displacement is decomposed on finite element (FE) shape functions [6].

The proposed method can be linked to data pruning or data cleaning methods described in the literature for machine learning [7]. The aim of these procedures are not to reduce data storage but to improve the data quality by accurate outliers detection for instance [8]. In [9], a data pruning method is employed to filter the noise in the dataset.

Using the FE approximation of the experimental fields paves the way to further simulations. In the calibration procedure, the full-field measurements are used as inputs of an inverse problem that aims to determine a given set of parameters $\boldsymbol{\mu} = \{\mu_1, \ldots, \mu_m\}$. These parameters are the coefficients of given constitutive equations. Their values are unknown or not known precisely. The most straightforward method is called Finite Element Model Updating (FEMU) (see Kavanagh and Clough [10] and Kavanagh [11]). It is a rather common way to optimize a set of parameters taking into account the experimental data and balance equations in mechanics. It consists in computing the discrepancy between the FE approximation of the experimental fields and the FE simulations. Thus, an optimization loop is done on $\boldsymbol{\mu}$ where the FE method is used as a tool for assessing the relevance of the parameter set. The objective function, or cost function, of the optimization can focus on the difference between the computed and experimental displacement fields (FEMU-U), forces (FEMU-F, or force balance method), or the strain fields (FEMU-$\varepsilon$) or a mix between all these sub-methods. A review of FEMU applications can be found in [12]. The method is particularly suitable for:

- Non-isotropic materials (e.g., materials having mechanical properties that depend on their orientation [13,14], such as the human skin [15]);
- Heterogeneous materials such as composites [16];
- Heterogeneous tests such as open-hole tests (e.g., [13,14]) or CT-samples [17];
- Special cases of local phenomena such as strain localization or necking (e.g., Forestier et al. [18], Giton et al. [19]) or the illustrating case of the present paper;
- Multi-materials configurations (e.g., solder joints studied in [20] or heterogeneous material identification done in [21]); and
- Determination of the boundary conditions [22].

One of the recent developments concerning FEMU is to couple this method with reduced order models (ROMs) to cut down the computation time in the parameters optimization loop. An example of such recent developments can be found in [23] where a method called FEMU-b is highlighted, or in [24]. The FEMU-b consists in determining an intermediate space of predominant empirical modes associated to a reduction procedure, such as the Proper Orthogonal Decomposition (see Aubry et al. [25]) or the Proper Generalized Decomposition (PGD) [26]. The discrepancy is computed between the experimental

and simulated reduced variables, where the reduced variables are solutions of reduced equations. In this paper, we show that the proposed data pruning method is consistent with a reduced order modeling of the equations to be calibrated. A FEMU-b is introduced, so we take into account the lack of experimental data due to the pruning procedure.

In [27], it has been shown that ROMs can be supplemented by a reduced integration domain (RID), by following a hyper-reduction method. In this method, a RID is a subdomain of a body, where the reduced equations are set up. In the proposed approach, we do not modify the cubature scheme involved in mechanical equations, as proposed by Hernandez et al. [28], but we restrict the cubature to a subdomain. This leads the way for data pruning methods that preserve calibration capabilities. Here, the dimensionality reduction of experimental data enables the restriction of experimental data to a RED. This RED is a subdomain of the specimen where the experimental data are sampled. It is not necessarily a connected domain. The flowchart of the proposed approach to data pruning is shown in Figure 1. After pruning, the data related to the domain occupied by a specimen, denoted by $\Omega$, are restricted to a RED denoted by $\Omega_R$. The way the model calibration is done, depends on the nature of the data available in a storage system. If the data are not pruned, then a conventional calibration by the FEMU method is possible. Otherwise, calibration by a FEMU-b method is recommended. In this paper, the calibration capabilities after data pruning are assessed by using the FEMU with an hybrid hyper-reduction method ($H^2$ROM) [29]. Hence, the FEMU-b is not done on the complete domain but on the RED determined by the data pruning. The result is a fast calibration procedure, with low memory requirement and a validated data pruning protocol. Contrary to usual hyper-reduction methods, the domain where the equations to be calibrated are setup is not generated by using simulation data. It derives from the data pruning procedure applied to experimental data.

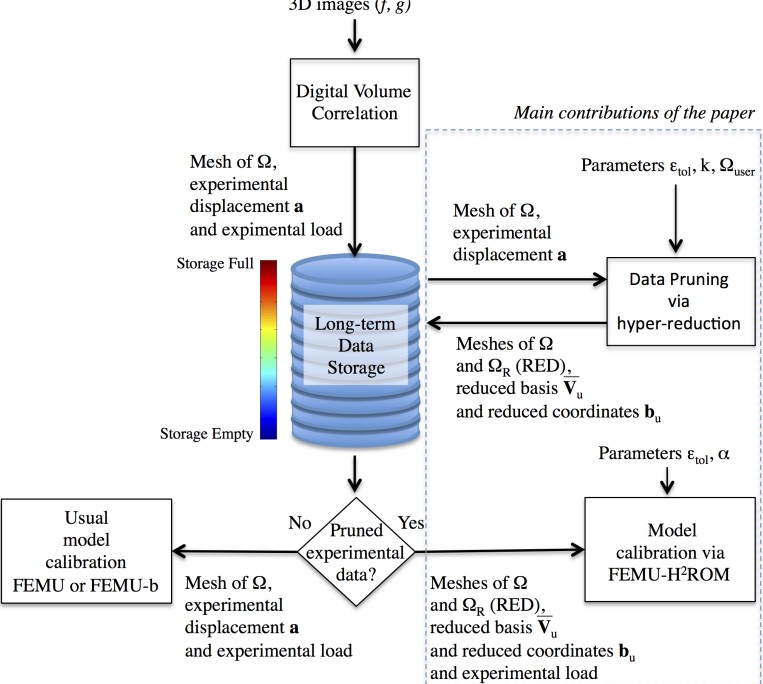

**Figure 1.** Pruning of experimental data related to DVC, via hyper-reduction. Calibration capabilities of constitutive equations are preserved after data pruning. The experimental data related to the domain occupied by a specimen, denoted by $\Omega$, are restricted to a reduced experimental domain denoted by $\Omega_R$. The way the model calibration is done, depends on the nature of the data available in the storage system.

The remaining part of the paper is structured as follows. In Section 3, the proposed method for data pruning is described. The DVC is recalled. A dimensionality reduction then hyper-reduction

are performed to compute the pruned data. The pruning procedure is applied in Section 4 on a resin-bonded sand tested in in situ uniaxial compression with X-ray tomography. In Section 5, the calibration of an elastoplastic model enables validating the pruning protocol. Details on the experimental data are available in the form of supplementary files. These data allow the proposed data pruning to be reproduced.

## 2. Notations

Second-order tensors are denoted by $\underset{\sim}{a}$. Matrices are denoted by capital bold letters **A** and vectors are denoted by bold lowercase characters **a**. The colon notation is used to denote the extraction of a submatrix or a vector (at column $i$ for example): $\mathbf{a} = \mathbf{A}[:, i]$. Sets of indices are denoted by calligraphic characters $\mathcal{A}$. The element of a matrix **A** at row $i$ and column $j$ is denoted $A_{ij}$ or $A_{\alpha}[i, j]$ when the matrix notation $\mathbf{A}_{\alpha}$ has a subscript. $\bar{\mathbf{a}}$ is the restriction of **a** to the reduced experimental domain.

## 3. Data Pruning by Following an Hyper-Reduction Scheme

In the proposed approach, the experimental displacements observed on the domain occupied by a specimen are restricted to the RED $\Omega_R$. The smaller is the extent of $\Omega_R$, the smaller is the memory requirement to store the pruned data. Without any constraint, the best memory saving is obtained by saving the parameters $\mu$ that best replicate the experimental data. In that situation, usual FEMU methods are sufficient. Here, the following constraint is taken into consideration. The data pruning should not prevent changes in the way constitutive equations are set up, as these equations may evolve in the future. Knowledge in mechanics is evolving and so are models. Thus, after the data have been pruned, the experimental data saved in the storage system must allow the calibration of constitutive equations. To ensure consistency between the computational complexity of the calibration procedure and the accuracy of the pruning data, we propose hyper-reduced equations for this calibration. In our opinion, it does not make sense to perform complex simulations during such a calibration with a poor representation of the experimental data.

### 3.1. Digital Volume Correlation

Let us consider a specimen occupying the domain $\Omega$ undergoing a certain mechanical test. With image acquisition techniques, grayscale images are obtained in 3D. The Digital Volume Correlation aims to determine the displacement field **u** at every position **x** in $\Omega$ at a given deformed state at time $t$. $f$ and $g$ are the gray levels at the reference and deformed states. They are related by the equation:

$$g(\mathbf{x}) = f(\mathbf{x} + \mathbf{u}(\mathbf{x}, t)) \tag{1}$$

The best displacement field is estimated via the minimization of the following residual:

$$\phi^2(\mathbf{u}, t) = \int_{\Omega} \left[ \mathbf{u}(\mathbf{x}, t).\nabla f(\mathbf{x}) + f(\mathbf{x}) - g(\mathbf{x}) \right]^2 d\mathbf{x} \tag{2}$$

where $\nabla f$ is the gradient of $f$. This is an ill-posed problem. To get a well-posed problem, the displacement field can be restricted to a kinematic subspace. Here, the displacement field is assumed to be decomposed over a set of vector functions $\boldsymbol{\psi}_j(\mathbf{x})$ that corresponds to the shape functions of a FE model defined on $\Omega$.

$$\mathbf{u}(\mathbf{x}, t) = \sum_{i=1}^{N_d} a_i(t) \boldsymbol{\psi}_i(\mathbf{x}) \tag{3}$$

where $N_d$ is the number of degrees of freedom of the mesh, $a_i$ the $i$th nodal degree of freedom in the FE model. **a** denotes the vector of degrees of freedom to be determined. With this restriction to the

kinematic subspace, the function $\phi$ is now a quadratic form of the $a_i$, and its minimization is a linear system, set up for each observation of a deformed state:

$$\mathbf{M}\mathbf{a} = \mathbf{h} \tag{4}$$

where the matrix $\mathbf{M}$ and the vector $\mathbf{h}$ are:

$$M_{ij} = \int_{\Omega} (\boldsymbol{\psi}_i(\mathbf{x}).\nabla f(\mathbf{x})) \left( \boldsymbol{\psi}_j(\mathbf{x}).\nabla f(\mathbf{x}) \right) \mathrm{d}x \tag{5}$$

$$h_i = \int_{\Omega} [g(\mathbf{x}) - f(\mathbf{x})] \, \boldsymbol{\psi}_i(\mathbf{x}).\nabla f(\mathbf{x}) \mathrm{d}x \tag{6}$$

In the sequel, $N_t$ observations of the specimen deformation at time instants $t_j$, $j = 1, \ldots, N_t$, are considered. The DVC gives access to the final correlated displacement field $\mathbf{u}(\mathbf{x}, t_j)$ for each observations, through the coefficient vector $\mathbf{a}(t_j)$. From the displacement field, a strain field $\underset{\sim}{\varepsilon}$ is extracted assuming small strains:

$$\underset{\sim}{\varepsilon} = \frac{1}{2} \left( \underset{\sim}{\nabla} \mathbf{u} + \underset{\sim}{\nabla} \mathbf{u}^T \right) \tag{7}$$

This strain is thus calculated at each Gauss point of the mesh used for the DVC. For pressure dependent or plastic materials, it can be convenient to subdivide the strain field in its deviatoric part and its hydrostatic part:

$$\underset{\sim}{\varepsilon} = \underset{\sim}{\varepsilon}^s + \underset{\sim}{\varepsilon}^v, \quad \text{with } \underset{\sim}{\varepsilon}^v = \mathrm{tr} \left( \underset{\sim}{\varepsilon} \right) \underset{\sim}{I} \tag{8}$$

where $\underset{\sim}{I}$ is the unit tensor.

It is worth noting that the pruning procedure only focuses on the displacement and not on the strain. It is considered that the strain can be computed in post-processing (thanks to Equation (7)) and are not worth saving. The strain tensor is actually considered as temporary data used to compute a reduced experimental domain.

### 3.2. Dimensionality Reduction

The first step of the pruning procedure consists in performing a dimensionality reduction of the experimental data. It is based on singular value decomposition. This approach is similar to the Principal Component Analysis (PCA). However, here, a reduced basis of empirical modes is obtained without centering the data.

The experimental data from DVC are saved into two matrices, $\mathbf{Q}_u$ and $\mathbf{Q}_\varepsilon$ defined as:

$$Q_u[i, j] = a_i(t_j), \ i = 1, \ldots, N_d, \ j = 1, \ldots, N_t \tag{9}$$

and

$$Q_\varepsilon[i, j] = \varepsilon^s_{\alpha\beta}(e_\gamma, t_j) \tag{10}$$

$$Q_\varepsilon[i, j + N_t] = \varepsilon^v_{\alpha\beta}(e_\gamma, t_j) \tag{11}$$

where $e_\gamma$ is the $\gamma$th Gauss point, and:

$$i = \beta + 3(\alpha - 1) + 9(\gamma - 1)$$
$$\alpha = 1, \ldots, 3, \ \beta = 1, \ldots, 3, \ \gamma = 1, \ldots, N_g$$
$$j = 1, \ldots, N_t$$

with $N_g$ being the number of integration points in the mesh. $\mathbf{Q}_u$ is a $N_d \times N_t$ matrix and $\mathbf{Q}_\varepsilon$ is a $(9N_g) \times (2N_t)$ matrix. For the sake of simplicity, we do not account for the symmetry of the strain tensor.

The first step of the pruning procedure consists in performing a first dimensionality reduction of the DVC data. Only the reduced basis and coordinate are kept instead of the snapshot matrix $\mathbf{Q}_u$. The procedure is also done on the snapshot matrix of the stain $\mathbf{Q}_\varepsilon$ but not in order to reduce storage (as the stain data are not saved). The corresponding reduced basis is used as a temporary tool to compute afterwards the reduced domain. The determination of the empirical modes is performed thanks to a Singular-Value Decomposition (SVD):

$$\mathbf{Q}_u = \mathbf{V}_u \mathbf{S}_u (\mathbf{W}_u)^T + \mathbf{R}_u \tag{12}$$

$$\mathbf{Q}_\varepsilon = \mathbf{V}_\varepsilon \mathbf{S}_\varepsilon (\mathbf{W}_\varepsilon)^T + \mathbf{R}_\varepsilon \tag{13}$$

where $\mathbf{V}_x \in \mathbb{R}^{N_d \times N_x}$, with $x = u$ or $\varepsilon$, is an empirical reduced basis for displacement or strain, respectively, $N_x \leq \text{rank}(\mathbf{Q}_x)$, $\mathbf{S}_x = \text{diag}(\sigma_{x1}, \ldots, \sigma_{xN_x}) \in \mathbb{R}^{N_x \times N_x}$, $\sigma_{x1} \geq \sigma_{x2} \geq, \ldots, \geq \sigma_{xN_x}$ and $\mathbf{W}_x \in \mathbb{R}^{N_t \times N_x}$. Both $\mathbf{V}_x$ and $\mathbf{W}_x$ are orthogonal. The residual $\mathbf{R}_x$ has a 2-norm such as:

$$\|\mathbf{R}_x\|_2 = \sigma_{x\, N_x+1} < \epsilon_{tol}\, \sigma_{x1}, \quad x = u \text{ or } \varepsilon \tag{14}$$

where $\epsilon_{tol}$ is a numerical parameter (typically, $10^{-3}$). According to the Eckart–Young theorem, the matrix $\mathbf{V}_x (\mathbf{V}_x)^T \mathbf{Q}_x$ is the best approximation of rank $N_x$ for $\mathbf{Q}_x$ by using the reduced basis $\mathbf{V}_x$.

The relevance of the dimensionality reduction of the displacement data appears to be conditioned by the difference between the number of time steps $N_t$ and the order of the approximation $N_u$, as $\mathbf{Q}_u \in \mathbb{R}^{N_d \times N_t}$ and $\mathbf{V}_u \in \mathbb{R}^{N_d \times N_u}$. In situ tests observed in X-ray CT tend to have few time steps so the first dimensionality reduction may not be efficient. Moreover, due to the resolution of the Computed Tomography, data have generally an important number of degrees of freedom. In other words, the snapshot matrix $\mathbf{Q}_u$ has a lot of lines ($N_d$) but few columns ($N_t$). The memory cost is mostly due to the number of dof of the problem. That is why the proposed pruning protocol is based on a hyper-reduction method in order to reduce significantly this number of dof.

### 3.3. Reduced Experimental Domain

The proposed pruning method has its roots in the hyper-reduction method [30]. We are not able to prove that the proposed approach has a strong physical basis for pruning data according to an appropriate metric. The proposed approach is heuristic, but it fulfills some mathematical properties. A hyper-reduced order model is a set of FE equations restricted to a RID when seeking an approximate solution of FE equations with a given reduced basis. In other words, this approach accounts for the low rank of the reduced approximation to set up the reduced equations of a given FE model. Let us denote by $\mathbf{a}^{FE} \in \mathbb{R}^{N_d}$ the solution of FE equations that aims to replicate the experimental vector $\mathbf{a}$, by using the same mesh. For a given reduced basis of rank $N_R$ $\mathbf{V} \in \mathbb{R}^{N_d \times N_R}$, the approximate reduced solution of the FE equations is denoted by $\mathbf{a}^R$ such that:

$$\mathbf{a}^R = \mathbf{V}\, \mathbf{b}^R \tag{15}$$

where $\mathbf{b}^R \in \mathbb{R}^{N_R}$ are the variables of the reduced order model. It turns out that the rank of the reduced FE equations must be $N_R$ in order to find a unique solution $\mathbf{b}^R$. Since $N_d$ is usually larger than $N_R$, few FE equations that preserve the rank of the reduced system exist. By following the hyper-reduction method proposed in [30], this selection is achieved by considering balance equations set up on a RID. In former works on hyper-reduction, the RID were generated by using simulation data.

Here, the RED is similar to a RID, but its construction uses solely experimental data, that is to say that the reduced basis used to perform this row selection comes from Equations (12) and (13). That is why the pruning method is called a model-free approach. One of the advantages of such method is that the data pruning does not have to be performed again if the constitutive model is changed. The RED is denoted by $\Omega_R \subset \Omega$.

In the hyper-reduction method, the RID is generated by the assembly of elements containing interpolation points related to various reduced bases. These reduced bases are usually extracted from simulation data generated by a given mechanical model for various parameter variations [30]. Here, the RED construction is based exclusively on the reduced bases related to $\mathbf{Q}_u$ and $\mathbf{Q}_\varepsilon$. The RED is the union of several subdomains: $\Omega_u$ and $\Omega_\varepsilon$ generated from the reduced matrices $\mathbf{V}_u$ and $\mathbf{V}_\varepsilon$, a domain denoted by $\Omega_+$ corresponding to a set of neighboring elements to the previous subdomains, and a zone of interest (ZOI) denoted by $\Omega_{user}$. In the sequel, $\Omega_{user}$ is set up to evaluate the force applied by the experimental setup on the specimen.

$\Omega_u$ is designed as if we would like to reconstruct experimental displacements outside $\Omega_u$ by using $\mathbf{V}_u$ and given experimental displacement in $\Omega_u$. On a restricted subdomain $\Omega_u$, we only have access to a restricted set of nodal displacements. The set of their indices is denoted by $\mathcal{P}_u$. The set of remaining displacement indices is denoted by $\mathcal{H}_u$ such that $\mathbf{a}[\mathcal{H}_u]$ is the vector to be reconstructed by knowing $\mathbf{a}[\mathcal{P}_u]$. Various approaches have been proposed in the literature to perform this kind of reconstruction. They are related to data completion [31] or data imputation [32] for instance. Here, we have the opportunity to choose the set $\mathcal{P}_u$, because the reconstruction issue is only formal. By using the DEIM method proposed in [33], we can obtain the set $\mathcal{P}_u$ such that $\mathbf{V}_u[\mathcal{P}_u,:]$ is a square and invertible matrix. Then, in that situation, the number of selected degrees of freedom in $\mathcal{P}_u$ is the number of empirical modes in $\mathbf{V}_u$. However, in the present application, this set could be too small to get robust calibrations after data pruning. Then, we propose a modification of the DEIM algorithm in order to multiply the number of selected indices by a given factor $k$. We name this algorithm $k$-Selection with empIrical Modes ($k$-SWIM). The modified algorithm is shown in Algorithm 1. When $k = 1$, this algorithm is exactly the same as the usual DEIM algorithm in [34]. The issue here is not to replicate experimental data via an interpolation scheme, but via calibrated FE simulations (by using $k > 1$). In the sequel, the set of selected indices by using k-SWIM is denoted by $\mathcal{P}_u^{(k)}$. The same reasoning is applied to the reconstruction of the experimental strain tensors. The k-SWIM algorithm applied to $\mathbf{V}_\varepsilon$ defines $\mathcal{P}_\varepsilon^{(k)}$.

For given sets of indices $\mathcal{P}_u^{(k)}$ and $\mathcal{P}_\varepsilon^{(k)}$, the RED is:

$$\Omega_R := \Omega_u \cup \Omega_\varepsilon \cup \Omega_+ \cup \Omega_{user}, \qquad \Omega_u := \cup_{j \in \mathcal{P}_u^{(k)}} \mathrm{supp}(\boldsymbol{\psi}_j) \qquad \Omega_\varepsilon := \cup_{j \in \mathcal{P}_\varepsilon^{(k)}} \mathrm{supp}(\boldsymbol{\psi}_j^\varepsilon). \tag{16}$$

where supp is the support of the function and $\boldsymbol{\psi}_j^\varepsilon$ are the shape functions related to the strain tensor in the FE model used to compute $\mathbf{a}$.

---

**Algorithm 1:** $k$-SWIM Selection of Variables with EmpIrical Modes

---

    **Input**   :integer $k$, linearly independent empirical modes $\mathbf{v}_l \in \mathbb{R}^d$, $l = 1, \dots, M$

    **Output:** variables index set $\mathcal{P}^{(k)}$

1   set $\mathcal{P}_0 := \varnothing$, $j = 0$, $\mathbf{U}_1 = [\,]$ ;                `// initialization`

2   **for** $l = 1, \dots, M$ **do**

3      $\mathbf{r}_l \leftarrow \mathbf{v}_l - \mathbf{U}_l \left( (\mathbf{U}_l[\mathcal{P}_j,:])^T \mathbf{U}_l[\mathcal{P}_j,:] \right)^{-1} (\mathbf{U}_l[\mathcal{P}_j,:])^T \mathbf{v}_l[\mathcal{P}_j]$ ;     `// residual vector`

4      **for** $n = 1, \dots, k$ **do**

5          $j \leftarrow j + 1$ ;             `// add the k largest value of the residual`

6          $i_j \leftarrow \arg\max_{i \in \{1,\dots,d\} \setminus \mathcal{P}_{j-1}} |\mathbf{r}_I[i]|$ ;         `// index selection`

7          $\mathbf{r}_l[i_j] \leftarrow 0$ ;             `// variable already selected`

8          $\mathcal{P}_j \leftarrow \mathcal{P}_{j-1} \cup \{i_j\}$ ;          `// extend index set`

9      $\mathbf{U}_{l+1} \leftarrow [\mathbf{v}_1, \dots, \mathbf{v}_l]$ ;           `// truncated reduced matrix`

10   set $\mathcal{P}^{(k)} := \mathcal{P}_j$.

---

Algorithm 1 is properly defined if in Line 3 the matrix $(\mathbf{U}_l[\mathcal{P}_j,:])^T \mathbf{U}_l[\mathcal{P}_j,:]$ is invertible, for $l > 1$ with $j = (l-1)k$, or equivalently if the following property is fulfilled.

**Theorem 1.** $\mathbf{U}_{l+1}[\mathcal{P}_{j+k},:]^T \mathbf{U}_{l+1}[\mathcal{P}_{j+k},:]$ *is invertible for* $l > 0$ *and* $j = (l-1)k$.

**Proof.** Let us assume that $\mathbf{U}_l[\mathcal{P}_j,:]^T \mathbf{U}_l[\mathcal{P}_j,:]$ is invertible for $l > 1$ and $j = (l-1)k$. Then, we compute $\mathbf{r}_l$. $(\mathbf{v}_l)_{l=1}^M$ is a set of linearly independent vectors. Thus, $\max_{i \in \{1,...,d\}} |\mathbf{r}_l[i]| > 0$. Let us introduce the first additional index, $j^\star = (l-1)k + 1$, $\mathcal{P}_{j^\star} = \mathcal{P}_j \cup \{\arg\max_{i \in \{1,...,d\}} |\mathbf{r}_l[i]|\}$ and the following residual vector:

$$\mathbf{r}_l^\star = \mathbf{v}_l[\mathcal{P}_{j^\star}] - \mathbf{U}_l[\mathcal{P}_{j^\star},:]\left((\mathbf{U}_l[\mathcal{P}_j,:])^T \mathbf{U}_l[\mathcal{P}_j,:]\right)^{-1} (\mathbf{U}_l[\mathcal{P}_j,:])^T \mathbf{v}_l[\mathcal{P}_j]$$

Then, $\mathbf{r}_l^\star = \mathbf{r}_l[\mathcal{P}_{j^\star}]$ and $\|\mathbf{r}_l^\star\|_2 > |\mathbf{r}_l[j^\star]| > 0$. Thus, $\mathbf{U}_{l+1}[\mathcal{P}_{j^\star},:]$ is full column rank. Since $\mathcal{P}_{j^\star} \subset \mathcal{P}_{j+k}$, then $\mathbf{U}_{l+1}[\mathcal{P}_{j+k},:]$ is full column rank and $\mathbf{U}_{l+1}[\mathcal{P}_{j+k},:]^T \mathbf{U}_{l+1}[\mathcal{P}_{j+k},:]$ is invertible. In addition, $\mathbf{U}_2[\mathcal{P}_k,:] = \mathbf{v}_1[\mathcal{P}_k]$ is a non-zero vector. Then, $\mathbf{U}_2[\mathcal{P}_k,:]^T \mathbf{U}_2[\mathcal{P}_k,:] > 0$ is invertible. $\square$

Another interesting property is the possible cancellation of the data pruning by using a large value of the parameter $k$ in the input of Algorithm 1. The following property holds.

**Theorem 2.** *If* $k = N_d$ *and if* $|\mathbf{V}_u[i,1]| > 0 \; \forall i = 1, \ldots, N_d$, *then* $\Omega_R = \Omega$. *The RED covers the full domain and all the data are preserved.*

**Proof.** By following Algorithm 1, for $l = 1$ with $k = N_d$ and $\mathbf{V}_u$ as inputs (in the algorithm, $d = N_d$), we obtain $\mathbf{q}_l = \mathbf{V}_u[:,1]$. If $|\mathbf{V}_u[i,1]| > 0 \; \forall i = 1, \ldots, N_d$, then $\mathcal{P}_k = \{1, \ldots, N_d\}$. Hence, $\mathcal{P}_u^{(N_d)} = \{1, \ldots, N_d\}$ and $\Omega_u = \Omega$ and $\Omega_R = \Omega$. $\square$

The second theorem is quite restrictive. In practice, large values of $k$, with $k < N_d$, enable preserving all the data. The value of $k$ has to be chosen according to the size of the memory that we would like the free up.

*3.4. Experimental Data Restricted to the RED*

For a given RED, $\Omega_R$, a set of selected degrees of freedom indices can be defined as:

$$\mathcal{F} = \left\{ i \in \{1, \ldots, N_d\} | \quad \int_{\Omega \setminus \Omega_R} \psi_i^2(\mathbf{x}) \mathrm{d}x = 0 \right\} \tag{17}$$

The degrees of freedom in $\mathcal{F}$ are not connected to elements outside $\Omega_R$. We denote by $\mathcal{I}$ the degrees of freedom on the interface between $\Omega_R$ and $\Omega \setminus \Omega_R$:

$$\mathcal{I} = \left\{ i \in \{1, \ldots, N_d\} | \quad i \notin \mathcal{F}, \int_{\Omega_R} \psi_i^2(\mathbf{x}) \mathrm{d}x > 0 \right\} \tag{18}$$

The union of these two set is denoted by $\overline{\mathcal{F}}$:

$$\overline{\mathcal{F}} = \mathcal{I} \cup \mathcal{F} \tag{19}$$

It contains all the indices of the degree of freedom in $\Omega_R$.

We denote by $\overline{\mathbf{Q}}_u \in \mathbb{R}^{\mathrm{card}(\overline{\mathcal{F}}) \times N_t}$ the experimental data restricted to the RED, such that:

$$\overline{\mathbf{Q}}_u = \mathbf{Q}_u[\overline{\mathcal{F}},:] \tag{20}$$

An additional SVD is performed on these experimental data such that:

$$\overline{\mathbf{Q}}_u = \overline{\mathbf{V}}_u \mathbf{S}_u'(\mathbf{W}_u')^T + \mathbf{R}_u' \tag{21}$$

$$\mathbf{b}_u(t_j) = (\overline{\mathbf{V}}_u)^T \mathbf{a}(t_j)[\overline{\mathcal{F}}], j = 1, \ldots, N_t \tag{22}$$

When the RED is available, the experimental data are restricted to $\Omega_R$ and the data to be stored are:

1.  The pruned reduced basis $\overline{\mathbf{V}}_u$, and the consecutive reduced coordinates $(\mathbf{b}_u(t_j))_{j=1,...,N_t}$.
2.  The full mesh of $\Omega$ and the mesh of $\Omega_R$ ($\mathcal{F}$ and $\mathcal{I}$).
3.  The load history applied to the specimen on the subdomain $\Omega_{user}$.
4.  Usual metadata related to the experiment (temperature, material parameters, etc.).

It is also advised to store the statistical distribution of a value of interest in the full domain and in the reduced domain. These data can be saved as histograms, for example. In this present paper, the shear strain distribution was saved, as this variable is extremely interesting in the case of strain localization. The additional memory cost is actually negligible as it consists in storing a few hundred floats.

The data concerning the strains are not stored as they can be computed with the displacement data thanks to Equation (7).

Generally, in-situ experiments observed in X-ray CT do not have numerous time steps, hence the above dimensionality reduction via SVD does not reduce drastically the size of the data to store. This is illustrated with the following example in Section 4. The hyper-reduction of the domain is actually the predominant step for data pruning.

### 3.5. Reduced Mechanical Equations Set Up on the Reduced Experimental Domain

Let us denote by $\mathbf{r}^{FE}$ the residual of the FE equations that have to be calibrated such that:

$$\mathbf{r}^{FE}(\mathbf{a}^{FE}) = 0 \tag{23}$$

For the sake of simplicity, we do not introduce the parameters $\boldsymbol{\mu}$ in the FE residual. Since the experimental data are restricted to the RED by following a hyper-reduced setting, the mechanical equations submitted to the calibration procedure are also hyper-reduced. We denote by $\bar{\mathbf{r}}^{FE}$ the partial computation of the FE residual restricted to the RED. $\bar{\mathbf{r}}^{FE}$ is the FE residual computed solely on a mesh of the RED. This mesh is termed reduced mesh. To account for the reduced mesh, a renumbering of the set $\mathcal{F}$, denoted by $\mathcal{F}^\star$, is defined such that:

$$\mathcal{F} = \overline{\mathcal{F}}[\mathcal{F}^\star] \tag{24}$$

where $\mathcal{F}^\star$ is the set of degrees of freedom related to the reduced mesh, that corresponds to $\mathcal{F}$ in the full mesh. They are located in blue squares in Figure 2b. Similarly, we define $\mathcal{I}^\star$ such that:

$$\mathcal{I} = \overline{\mathcal{F}}[\mathcal{I}^\star] \tag{25}$$

where $\mathcal{I}^\star$ is the set of degrees of freedom related to the reduced mesh that belongs to the interface between the RED and the remaining part of the full domain. The various sets of degrees of freedom are shown in Figure 2.

We assume that:

$$\bar{\mathbf{r}}^{FE}(\mathbf{a}'[\overline{F}])[\mathcal{F}^\star] = \mathbf{r}^{FE}(\mathbf{a}')[\mathcal{F}] \quad \forall \mathbf{a}' \in \mathbb{R}^{N_d} \tag{26}$$

This assumption means that the FE residuals at lines selected by $\mathcal{F}$, for any prediction $\mathbf{a}'$, can be computed on the reduced mesh, where the residuals depend only on degrees of freedom in $\overline{F}$. It is relevant in mechanical problems without contact condition, in the framework of first strain-gradient theory. We refer the reader to [35] for the extension of the hyper-reduction method to contact problems.

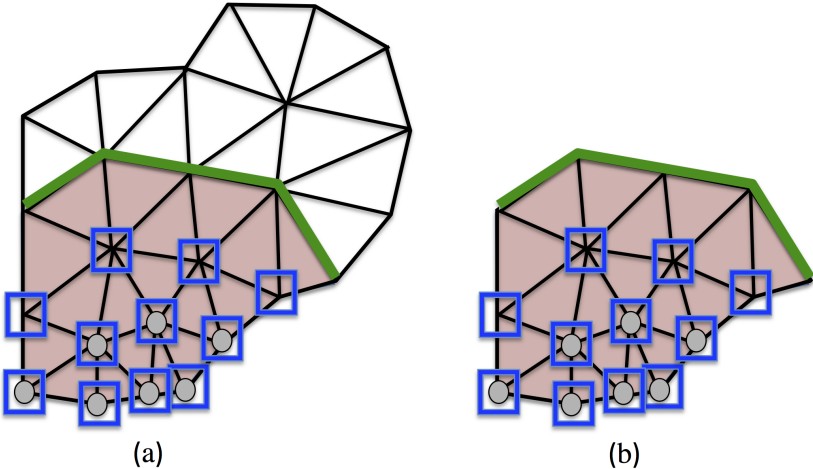

**Figure 2.** Schematic view of the reduced experimental domain, with linear triangular elements. In both figures, $\Omega_R$ is red. On the left, there is the mesh of $\Omega$. On the right, there is the reduced mesh (i.e., the mesh of $\Omega_R$ only). In (**a**), the nodes having their degrees of freedom in $\mathcal{I}$ are on the green line, the nodes having their degrees of freedom in $\mathcal{F}$ are in blue squares, and the grey nodes have their degrees of freedom in $\mathcal{R}$. In (**b**), the green line, the blue squares and the grey nodes are related to $\mathcal{I}^\star$, $\mathcal{F}^\star$ and $\mathcal{R}^\star$, respectively.

Both simulation data and experimental data are incorporated in a reduced basis dedicated to the calibration procedure, after the pruning of the experimental data. In the sequel, this reduced basis is extracted from data restrained to the RED, by using the SVD. Let us denote by $\mathbf{X}$ all the data available on the full mesh. Then, after the restriction of data to the reduced mesh, the empirical reduced basis is related to $\overline{\mathbf{X}} = \mathbf{X}[\overline{\mathcal{F}}, :]$:

$$\overline{\mathbf{X}} = \overline{\mathbf{V}}\,\overline{\mathbf{S}}\,\overline{\mathbf{W}}^T + \overline{\mathbf{R}}, \ \overline{\mathbf{V}} \in \mathbb{R}^{\text{card}(\overline{\mathcal{F}}) \times N_R} \tag{27}$$

with $\|\overline{\mathbf{R}}\| < \epsilon_{tol}\,\max(\text{diag}(\overline{\mathbf{S}}))$. $\overline{\mathbf{V}}$ is not a submatrix of a given $\mathbf{V}$. The way $\mathbf{X}$ contains both simulation data and experimental data is user dependent. In the proposed example, we are using a derivative extended proper orthogonal decomposition (see Schmidt et al. [36]) as explained in Section 5.2.

When the reduced basis contains empirical modes and few FE shape functions located in $\Omega_R$, the method is termed hybrid hyper-reduction [29,35]. The hybrid FE/reduced approximation is obtained by adding few columns of the identity matrix to $\overline{\mathbf{V}}$. In this hybrid approximation, we only add FE degrees of freedom that are not connected to the degrees of freedom in $\mathcal{I}^\star$. The resulting set of degrees of freedom is denoted by $\mathcal{R}^\star$ (see Figure 2). In [29] it has been shown that this permits to have strong coupling in the resulting hybrid approximation. Let us define the subdomain connected to $\mathcal{I}$:

$$\Omega_{\mathcal{I}} = \cup_{i \in \mathcal{I}}\text{supp}(\boldsymbol{\psi}_i) \cap \Omega_R \tag{28}$$

Then, we get:

$$\mathcal{R} = \left\{ i \in \overline{\mathcal{F}} \middle| \int_{\Omega_I} \boldsymbol{\psi}_i^2(\mathbf{x})\mathrm{d}x = 0 \right\} \tag{29}$$

and $\mathcal{R}^\star$ is such that:

$$\mathcal{R} = \overline{\mathcal{F}}[\mathcal{R}^\star] \tag{30}$$

The hybrid reduced basis is denoted by $\overline{\mathbf{V}}^H$. It reads, by using the Kronecker delta ($\delta_{ji}$):

$$\overline{\mathbf{V}}^H[:, 1:N_R] = \overline{\mathbf{V}}, \quad \overline{\mathbf{V}}^H[i, N_R + k] = \delta_{\mathcal{R}_k^\star\, i} \quad k = 1, \ldots, \text{card}(\mathcal{R}) \tag{31}$$

The equations of the hybrid hyper-reduced order model (H$^2$ROM) [35] reads: find $\mathbf{b}^H \in \mathbb{R}^{N_R+\mathrm{card}(\mathcal{R})}$ such that,

$$(\overline{\mathbf{V}}^H[\mathcal{F}^\star,:])^T\, \overline{\mathbf{r}}^{FE}(\overline{\mathbf{V}}^H\, \mathbf{b}^H)[\mathcal{F}^\star] = 0 \tag{32}$$

If the reduced equations do not have a full rank, it is suggested to remove the columns of $\overline{\mathbf{V}}$, in $\overline{\mathbf{V}}^H$, that cause the rank deficiency. When using the SVD to obtain $\overline{\mathbf{V}}$ from data, the last columns have the smallest contribution in the data approximation. These columns must be removed first in case of rank deficiency.

**Theorem 3.** *When $\Omega_R = \Omega$, then the hybrid hyper-reduced equations are the original FE equations on the full mesh.*

**Proof.** If $\Omega_R = \Omega$, then $\mathcal{I} = \varnothing$, $\mathcal{F}^\star = \mathcal{R}^\star = \{1,\ldots,N_d\}$ and the reduced mesh is the original mesh. In addition, all the empirical modes have to be removed from $\overline{\mathbf{V}}^H$ to get a full rank system of equations. Hence, $\overline{\mathbf{V}}^H$ is the identity matrix. Thus, the hybrid hyper-reduced equation are exactly the original FE equations. There is no complexity reduction. □

**Theorem 4.** *If $\epsilon_{tol}$ is set to zero; if both hybrid hyper-reduced equations and FE equations have unique solutions; if the FE solution $\mathbf{a}^{FE}$ belongs to the subspace spanned by the data $\mathbf{X}$; and if there exists a matrix $\mathbf{G}$ such that $\|\mathbf{a}^{FE} - \mathbf{X}\,\mathbf{G}\,\mathbf{a}^{FE}[\overline{\mathcal{F}}]\| = 0$ (i.e., the FE solution can be reconstructed by using the FE solution restricted to the RED), with $\mathbf{G} = \overline{\mathbf{W}}\,\overline{\mathbf{S}}^{-1}\,\overline{\mathbf{V}}^T$, then $\mathbf{b}^H[1 : N_R] = \overline{\mathbf{V}}^T\,\mathbf{a}^{FE}[\overline{\mathcal{F}}]$ and $\mathbf{b}^H[1 + N_R : card(\mathcal{R}) + N_R] = \mathbf{0}_\mathcal{R}^T$, where $\mathbf{0}_\mathcal{R}$ is a vector of zero in $\mathbb{R}^{card(\mathcal{R})}$. This means that the hyper-reduced solution is exact and the FE correction in the hybrid approximation is null.*

**Proof.** Let us introduce the matrix $\widehat{\mathbf{V}} = \mathbf{X}\,\overline{\mathbf{W}}\,\overline{\mathbf{S}}^{-1}$. Then,

$$\widehat{\mathbf{V}}[\overline{\mathcal{F}},:] = \overline{\mathbf{X}}\,\overline{\mathbf{W}}\,\overline{\mathbf{S}}^{-1} = \overline{\mathbf{V}} \tag{33}$$

If $\|\mathbf{a}^{FE} - \mathbf{X}\,\mathbf{G}\,\mathbf{a}^{FE}[\overline{\mathcal{F}}]\| = 0$ with $\mathbf{G} = \overline{\mathbf{W}}\,\overline{\mathbf{S}}^{-1}\,\overline{\mathbf{V}}^T$, so $\|\mathbf{a}^{FE} - \widehat{\mathbf{V}}\,\overline{\mathbf{V}}^T\,\mathbf{a}^{FE}[\overline{\mathcal{F}}]\| = 0$, then $\|\mathbf{a}^{FE}[\overline{\mathcal{F}}] - \overline{\mathbf{V}}\,\widehat{\mathbf{b}}^{FE}\| = 0$ with $\widehat{\mathbf{b}}^{FE} = \overline{\mathbf{V}}^T\,\mathbf{a}^{FE}[\overline{\mathcal{F}}]$ and $[(\widehat{\mathbf{b}}^{FE})^T, \mathbf{0}_\mathcal{R}^T]^T$ fulfills the following equation:

$$\overline{\mathbf{r}}^{FE}(\overline{\mathbf{V}}^H\,[(\widehat{\mathbf{b}}^{FE})^T, \mathbf{0}_\mathcal{R}^T]^T)[\mathcal{F}^\star,:] = 0 \tag{34}$$

Then, the balance equations of the hybrid hyper-reduced equations are fulfilled by $[(\widehat{\mathbf{b}}^{FE})^T, \mathbf{0}_\mathcal{R}^T]^T$. If both hybrid hyper-reduced equations and FE equations have unique solutions, then the solution of the hybrid hyper-reduced equations is $\mathbf{b}^H = [(\widehat{\mathbf{b}}^{FE})^T, \mathbf{0}_\mathcal{R}^T]^T$. □

## 4. Illustrating Example: Polyurethane Bonded Sand Studied with X-ray CT

### 4.1. Material and Test Description

The material studied here is a polyurethane bonded sand used in casting foundry to mold the internal cavities of foundry parts. The resin makes bonds between grains and improves drastically the mechanical properties of the cores (stiffness, maximum yield stress, traction strength, etc.). The material has been extensively studied with standards laboratory tests, focusing on macroscopic displacement-force curves. This casting sand was experimentally investigated by Jomaa et al. [37], Bargaoui et al. [38]. These macroscopic data are completed with an in-situ uniaxial compression test studied in X-ray CT on an as-received sample. According to Bargaoui et al. [38], the process used to make the cores (cold box process) guarantees the homogeneity of the material. In the sequel, the resin bonded sand is supposed homogeneous.

The sample is a parallelepiped ($20.0 \times 22.4 \times 22.5$ mm$^3$). The load was increased (with a constant displacement rate of 0.5 mm/min) and the displacement was stopped at several levels, noted $P_i$. During these stopped displacement periods, the sample was scanned with a tension beam of 80 kV and an intensity of 280 μA. $P_0$ corresponds to the initial state, before the appliance of the load. Then, seven tomography scans were performed at increasing compressed states. At $P_7$, the sample is broken. The bottom and top extremities were excluded from the images because of the artifacts induced by the plates. A grayscale image of the tested cemented sand is displayed in Figure 3. During the test, the reaction is measured at the top of the sample. It is plotted in Figure 4. The first six steps (non-broken sample) are situated before the peak of the loading curve.

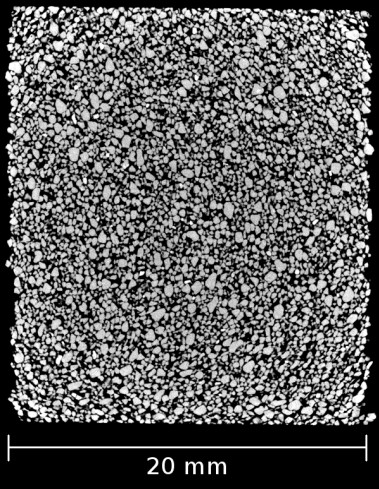

**Figure 3.** View of the sand.

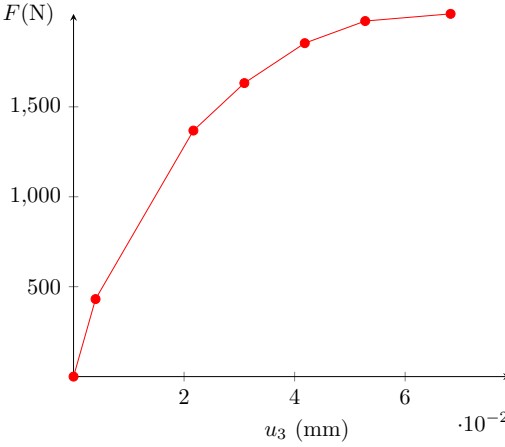

**Figure 4.** Measured top reaction.

### 4.2. DVC and Error Estimation

The displacement fields at these different stages were calculated using a 3D-digital image correlation (DVC) software named Ufreckles, developed by LaMCos (see [6]). A finite element continuum method is used to calculate the displacement field with a non-linear least square error minimization method. The chosen element size is near 0.5 mm. The final region of interest is $20.0 \times 22.4 \times 15.8$ mm$^3$. The top of the sample has been excluded. The DVC is performed on a parallelipedic mesh composed of around 470,000 degrees of freedom.

The DVC showed that the pre-peak displacement field is extremely non-homogeneous, as shown in Figure 5. The test showed a complex and rich behavior of the material tested with a non-homogeneous

displacement field and pre-peak strain bifurcations. The experimental data are very suited for testing the ability of a given model to predict such phenomena.

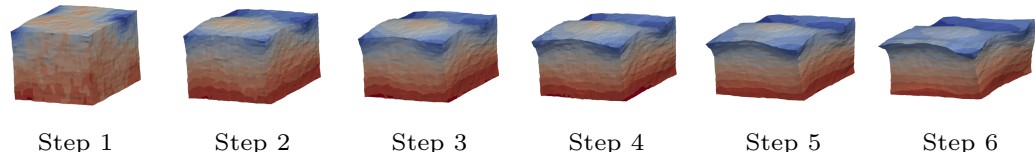

| Step 1 | Step 2 | Step 3 | Step 4 | Step 5 | Step 6 |

**Figure 5.** Magnitude of the experimental displacements at the pre-peak steps (deformed × 75).

### 4.3. Building the Reduced Experimental Basis

For a precise data pruning procedure, the experimental displacement and strain snapshot matrices are computed. The attention is drawn to the fact that the studied test does not have many time steps ($N_t = 7$) and the experimental mesh is not that big. The DVC matrices $\mathbf{Q}_u$ and $\mathbf{Q}_\varepsilon$ are, respectively, 474,405 × 7 and 1,774,080 × 14. If the truncated SVD is applied on these matrices, only six modes are extracted for the displacement and 13 for strain. As the number of time steps is rather small, the use of empirical modes does not reduce the size of the experimental data, as stated before.

In other words, the experimental data are not suited for the dimensionality reduction. This method is efficient on matrices with numerous columns and rather few lines, whereas tomographic data tend to have the exact opposite: few columns (time steps) and a lot of lines (degrees of freedom).

### 4.4. RED after DVC on the Specimen

During the test, the loading curve was measured at the top of the sample. To compare computed and measured reactions for model assessment, the elements at the top of the mesh are considered as a ZOI. In the remaining, $\Omega_+$ is one layer of elements around $\Omega_u \cup \Omega_\varepsilon \cup \Omega_{user}$.

The RED was determined varying the number $k$ of selected lines in the k-SWIM Algorithm. Its influence is assessed in Figure 6. For $k = 1$, the standard DEIM algorithm selects very few degrees of freedom. Most of the RED is actually the ZOI. This is due to the relatively low number of modes contained in the reduced basis (only 6). This apparent issue can be overcome by selecting more lines during the $k$-SWIM algorithm. When increasing $k$, the number of degrees of freedom linearly rises. The attention is drawn on the fact that the resultant RED for $k = 25$ or $k = 50$ are discontinuous, as is usually the case when using hyper-reduction methods. The newly selected zones are situated in the sheared regions. For the sake of reproducibility, the binary files related to $\mathbf{V}_u$, $\mathbf{b}_u$ and $\mathcal{P}_u$ are available as supplementary files.

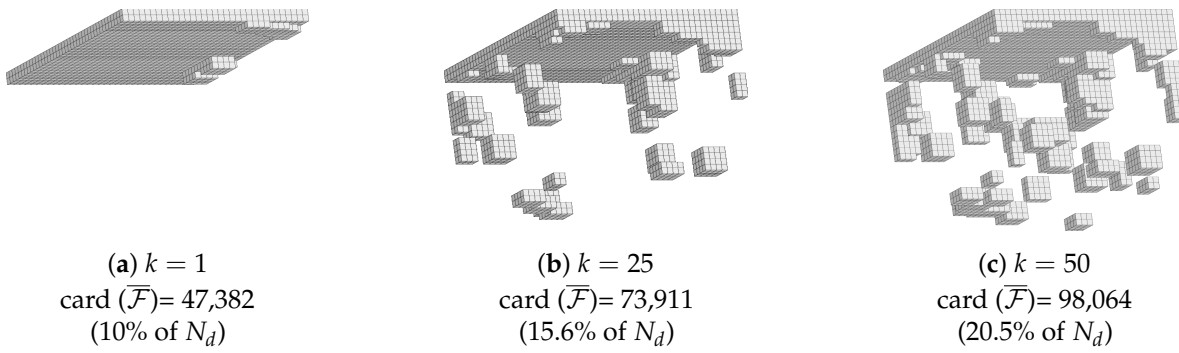

| **(a)** $k = 1$ | **(b)** $k = 25$ | **(c)** $k = 50$ |
| card ($\overline{\mathcal{F}}$)= 47,382 | card ($\overline{\mathcal{F}}$)= 73,911 | card ($\overline{\mathcal{F}}$)= 98,064 |
| (10% of $N_d$) | (15.6% of $N_d$) | (20.5% of $N_d$) |

**Figure 6.** Influence of $k$ in the $k$-SWIM algorithm.

The final RED was arbitrarily selected with $k = 25$ (around 15.6% of the total domain $\Omega$). It is displayed in Figure 6b. The reduced domain construction is analyzed in Figure 7 where the subdomains $\Omega_u$, $\Omega_\varepsilon$, $\Omega_{user}$ and $\Omega_I$ are displayed.

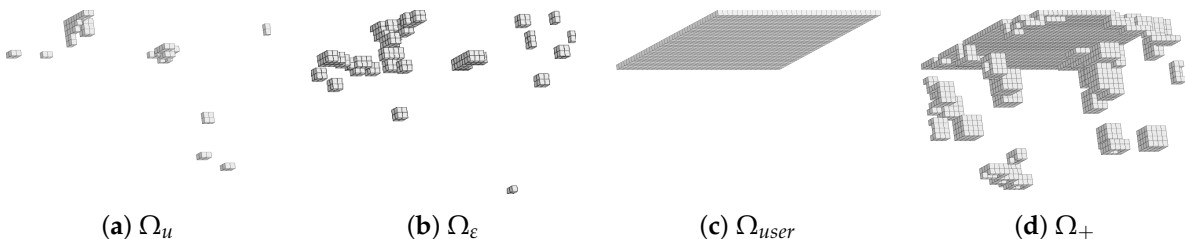

**(a)** $\Omega_u$       **(b)** $\Omega_\varepsilon$       **(c)** $\Omega_{user}$       **(d)** $\Omega_+$

**Figure 7.** Different subdomains for the selected RED for $k = 25$.

A summary of the different matrix sizes at each step is displayed in Table 1. As stated before, it is clear that for this kind of data, the PCA analysis does not reduce significantly the memory usage. The hyper-reduction scheme used allowed saving up to 85% of the memory space for the illustrating example.

**Table 1.** Size of the matrix stored at each step of the data pruning.

| Experimental Data | | Empirical Modes | | Pruned Data | |
|---|---|---|---|---|---|
| $\mathbf{Q}_u$ | $474{,}405 \times 7$ | $\mathbf{V}_u$ | $474{,}405 \times 6$ | $\overline{\mathbf{V}}_u$ | $73{,}911 \times 6$ |
| | | $\mathbf{b}_u$ | $6 \times 7$ | $\mathbf{b}_u$ | $6 \times 7$ |
| Memory Saved | | 15% | | 85% | |

## 5. Assessing the Relevance of the Pruned Data via Finite Element Model Updating-H²ROM

In this section, the relevance of the pruned data for further usage is discussed. The experimental data extracted from computed tomography can have various purposes. This paper focuses on its use for model calibration, and is illustrated with the in-situ compressive test of a resin bonded sand presented in the previous section. The main aim of this part is to prove that the RED computed thanks to a model free procedure is relevant to assess or calibrate an arbitrary constitutive model.

The model used for the illustrating example is a constitutive elastoplastic model with $m$ unknown parameters to calibrate. The procedure employed is a Finite Element Model Updating (FEMU) technique, coupled with an hybrid hyper-reduction method for the solution of approximate balance equations. The use of such method is straightforward as the input data are actually hyper-reduced. This approach is termed FEMU-H²ROM.

The FEMU-H²ROM method is resumed in the flowchart in Figure 8. The FEMU-H²ROM aims to find the best parameter $\boldsymbol{\mu}^*$ that replicate the experimental data available on the RED by using hyper-reduced equations. During the optimization procedure, the parameters are updated via hybrid hyper-reduced simulations. After few adaptation steps, the optimality of the parameter is checked by using a full FE simulation. If required, the reduced basis involved in the hyper-reduced simulation are updated.

### 5.1. Constitutive Model MC-CASM

#### 5.1.1. Presentation

The resin-bonded sand behavior is modeled with a relatively simple constitutive model based on the Cemented Clay and Sand Model (C-CASM). It consists in the extension of the Clay And Sand Model developed by Yu [39] for unbonded sand and clay to bonded geomaterials within the framework developed by Gens and Nova [40]. The C-CASM has been extensively described in [41]. The Modified Cemented Clay And Sand Model (MC-CASM) presented here has some modifications of the C-CASM:

- Addition of a damage law whose equation is phenomenological (based on cycled compressive tests).

- The hardening law of the bonding parameter $b$ is different: A first hardening precedes the softening. It is supposed here that the polyurethane resin goes through a first hardening before breaking.

It is supposed here that the yield function was previously calibrated with standard laboratory tests. The calibration concerns the parameters involved in the different damage and hardening laws that can be more difficult to assess with macroscopic loading curves. In the continuation of the paper, the equivalent von Mises stress is denoted $q$ and the mean pressure $p$. The MC-CASM equations are summarized hereafter.

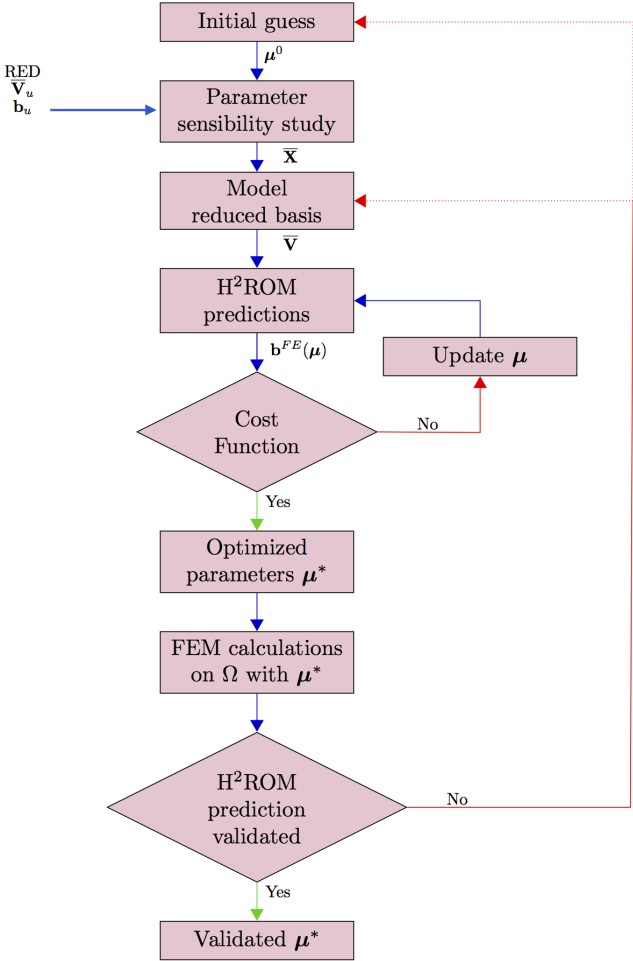

**Figure 8.** Flowchart of the FEMU-H$^2$ROM.

5.1.2. Yield Function and Plastic Flow

The yield function, $f$, of the constitutive model is defined by:

$$f(\sigma; p_c, b) = \left( \frac{q}{M(p + p_t)} \right)^n + \frac{1}{\ln r} \ln \left( \frac{p + p_t}{p_c(1+b) + p_t} \right) \tag{35}$$

where $M$, $r$, and $n$ are constant parameters that control the shape of the yield function. $p_c$ is the preconsolidation pressure, that is to say the maximum yield pressure during an isotropic compressive test (see Roscoe et al. [42]). $b$ is the bounding parameter modeling the amplification of the yield surface due to intergranular bonding. $p_t$ is the traction resistance of the soil defined by Gens and Nova [40] as:

$$p_t = \alpha b p_c \tag{36}$$

where $\alpha$ is a constant parameter modeling the influence of the binder on the traction resistance. The yield function is supposed to be calibrated. This means that $M$, $r$, $n$, $\alpha$ and the initial values of $p_c$ and $b$ are known. The yield surfaces of the unbonded (blue) and bonded sand (red) are plotted in Figure 9.

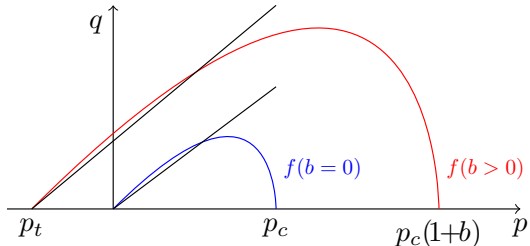

**Figure 9.** Yield surfaces in the $(p, q)$ plane.

### 5.1.3. Hardening and Damage Laws

The model has two hardening variables: the preconsolidation pressure $p_c$ and the bonding parameter $b$. The evolution of $p_c$ is directly controlled by the incremental plastic volumetric strain $\dot{\varepsilon}_v^p$, whereas $b$ relies on a plastic strain damage measure $h$:

$$\frac{\dot{p}_c}{p_c} = \mu_1 \dot{\varepsilon}_v^p \tag{37}$$

$$\dot{b} = (-be^{-h} + \mu_6 \mu_7 e^{-\mu_7 h})\dot{h} \tag{38}$$

The incremental value of $h$ is defined as a weighting of the effects of the incremental plastic shear strain and the incremental plastic volumetric strain:

$$\dot{h} = \mu_2 |\dot{\varepsilon}_s^p| + \mu_3 |\dot{\varepsilon}_v^p| \tag{39}$$

The model also includes a damage law whose formulation is purely phenomenological:

$$E = E_0(1 - D) \tag{40}$$

$$D = \mu_4 h^{\mu_5} \tag{41}$$

The hardening and damage laws provide $m = 7$ unknown parameters to calibrate.

### 5.2. Calibration Protocol by Using the Hybrid Hyper-Reduction Method

The FEMU-H$^2$ROM is preceded by an off-line phase similar to an unsupervised machine learning phase. It consists in building the empirical reduced basis $\overline{\mathbf{V}}$ that is mandatory to set up the hybrid hyper-reduced equations. It is similar to the first step of the data pruning method: a snapshot matrix is constructed based on simulations and experimental results (and not on experiments only).

The starting point of the off-line phase is to assess the parameter sensibilities of the model starting from an initial guess $\boldsymbol{\mu}^0 = \{\mu_1^0, \ldots, \mu_m^0\}$. This guess can come from a previous calibration, or a calibration done using macroscopic force–displacement curves of standard tests without predicting strain localization.

The off-line calculations are performed on the full domain $\Omega$ and thus can be time consuming. The boundary conditions are the experimental displacements taken from the computed tomography imposed at the top and the bottom of the sample. The displacement field is not imposed inside the sample because one of the aims of the model is to correctly capture the strain localization appearing inside the sample during the test, under the constraint of balance equations. Imposing the displacement field inside the specimen gives less balance equation to fulfill. $m$ calculations are made on $\Omega$. Attention

is drawn to the fact that these calculations can be done in parallel. Only the displacement snapshot matrices are needed. A total of $m + 1$ independent calculations are performed:

- One initial calculation where $\boldsymbol{\mu} = \boldsymbol{\mu}^0$, which gives $\mathbf{Q}_u^{FE}(\boldsymbol{\mu}^0)$;
- $m$ parameters sensibility calculations where $\boldsymbol{\mu} = \boldsymbol{\mu}^i = \{\mu_1^0, \ldots, \mu_i^0 + \delta\mu_i^0, \ldots, \mu_m^0\}$, which give $\mathbf{Q}_u^{FE}(\boldsymbol{\mu}^i)$ for $i = 1, \ldots, m$

Once done, these calculations are restricted to the reduced experimental domain $\Omega_R$. They are denoted $\overline{\mathbf{Q}}_u^{FE}(\boldsymbol{\mu}^i)$ for $i = 0, \ldots, m$. All these results have to be aggregated in one snapshot matrix $\overline{\mathbf{X}}$ before the computation of the empirical modes $\overline{\mathbf{V}}$. Instead of concatenating the $m + 1$ matrices into one, a DEPOD method is used (see Schmidt et al. [36]). This approach has been validated in previous works on model calibration with hyper-reduction (see Ryckelynck and Missoum Benziane [43]). This allows capturing the effects of each parameter variation.

$$\overline{\mathbf{X}} = [\alpha\overline{\mathbf{V}}_u\mathbf{b}_u, \ \overline{\mathbf{Q}}_u^{FE}(\boldsymbol{\mu}^0), \ \frac{\|\overline{\mathbf{Q}}_u^{FE}(\boldsymbol{\mu}^0)\|_F}{2\|\overline{\mathbf{Q}}_u^{FE}(\boldsymbol{\mu}^1) - \overline{\mathbf{Q}}_u^{FE}(\boldsymbol{\mu}^0)\|_F} \ (\overline{\mathbf{Q}}_u^{FE}(\boldsymbol{\mu}^1) - \overline{\mathbf{Q}}_u^{FE}(\boldsymbol{\mu}^0)), \ldots,$$

$$\frac{\|\overline{\mathbf{Q}}_u^{FE}(\boldsymbol{\mu}^0)\|_F}{2\|\overline{\mathbf{Q}}_u^{FE}(\boldsymbol{\mu}^m) - \overline{\mathbf{Q}}_u^{FE}(\boldsymbol{\mu}^0)\|_F} \ (\overline{\mathbf{Q}}_u^{FE}(\boldsymbol{\mu}^m) - \overline{\mathbf{Q}}_u^{FE}(\boldsymbol{\mu}^0))] \quad (42)$$

where $\|\cdot\|_F$ is the Frobenius norm. The first term $\alpha\overline{\mathbf{V}}_u\mathbf{b}_u$ corresponds to the pruned experimental data. It is weighted by a custom parameter $\alpha$ that enables giving more impact to the experimental fluctuations in the empirical modes. The finite element methods tends to smooth these fluctuations, thus provoking a certain loss of information.

Empirical modes depending on the factor $\alpha$ are displayed in Figure 10. For $\alpha = 0$, that is to say without experimental data in the bulk, the empirical modes have strong fluctuations only at the top and the bottom of the specimen, where the experimental boundary conditions are imposed. This can be explained by the natural smoothing that ensures the finite element method with rather elliptic equations. Increasing the importance of the experimental data tends to naturally perturb the displacement field inside the sample. Even for strongly perturbed modes ($\alpha = 10$), the last empirical mode is roughly smooth: this is due to the POD algorithm that filters the data. In the sequel, we choose $\alpha = 1$. The experimental data are as important as simulation data related to FE balance equations.

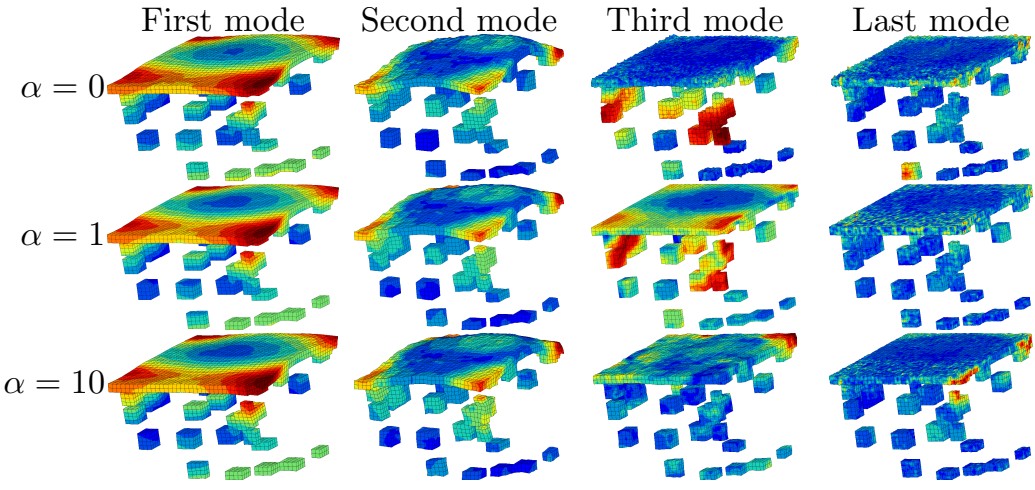

**Figure 10.** Magnitude of the displacement ($\sqrt{u_1^2 + u_2^2 + u_3^2}$) for each DEPOD mode depending on $\alpha$.

Once $\overline{\mathbf{V}}$ is available, the hybrid reduced basis $\overline{\mathbf{V}}^H$ can be defined. Then, the experimental reduced coordinates are projected on the empirical reduced basis to be compared during the optimization loop:

$$\mathbf{b}_u^H = (\overline{\mathbf{V}}^H)^T \, \overline{\mathbf{V}}_u \, \mathbf{b}_u \quad (43)$$

For the proposed example, there is a fast decay of the singular value (see Figure 11 where $\epsilon_{POD}$ is set to $10^{-4}$). When this decay is not sufficient to provide a small number of empirical modes, we refer the reader to [44–46] to cluster the data in order to divide the time interval and construct local reduced basis in time.

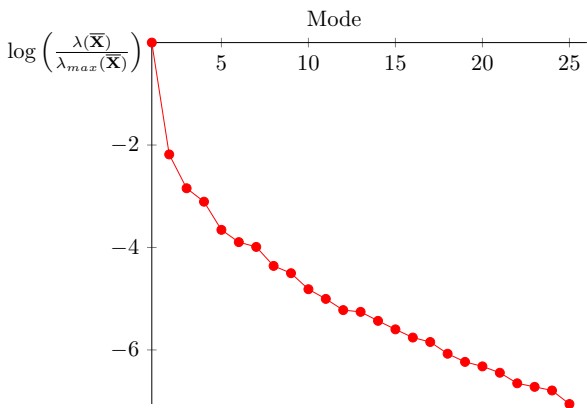

**Figure 11.** Singular values of $\overline{\mathbf{X}}$ verifying $\lambda(\overline{\mathbf{X}}) > \epsilon_{POD}\lambda_{max}(\overline{\mathbf{X}})$.

*5.3. Discussion on Dirichlet Boundary Conditions*

After the data pruning, experimental data are available in all $\Omega_R$. When displacements are constrained to follow the experimental data, we loose FE balance equations. The following theorem helps to discuss the Dirichlet boundary conditions.

**Theorem 5.** *If $\alpha > 0$, $\epsilon_{tol} = 0$; if the experimental data $\overline{\mathbf{Q}}_u = \overline{\mathbf{V}}_u\,\mathbf{b}_u$ fulfill the FE equations on $\Omega_R$ with the following additional Dirichlet boundary conditions:*

$$\mathbf{a}^{FE}(t_j,\boldsymbol{\mu})[\mathcal{I}] = \overline{\mathbf{Q}}_u[\mathcal{I}^*, j]; \tag{44}$$

*and if both hybrid hyper-reduced equations and FE equations on $\Omega_R$ are unique, then the solution of the hybrid hyper-reduced equation is the exact projection of the experimental data on the empirical reduced basis $\mathbf{b}^H(\boldsymbol{\mu}) = [(\overline{\mathbf{V}}^T\,\overline{\mathbf{Q}}_u)^T,\, \mathbf{0}_{\mathcal{R}}^T]^T$, with $\|\overline{\mathbf{Q}}_u - \overline{\mathbf{V}}\,\overline{\mathbf{V}}^T\,\overline{\mathbf{Q}}_u\| = 0$.*

**Proof.** If the solution of the FE equations in $\Omega_R$ is unique with Dirichlet boundary conditions on $\mathcal{I}^*$ equal to $\mathbf{a}^{FE}(t_j,\boldsymbol{\mu})[\mathcal{I}]$, then this solution is $\mathbf{a}^{FE}(t_j,\boldsymbol{\mu})[\overline{\mathcal{F}}]$. If $\overline{\mathbf{Q}}_u$ fulfills the FE equations on $\Omega_R$, with the additional Dirichlet boundary conditions, then:

$$\mathbf{a}^{FE}(t_j,\boldsymbol{\mu})[\overline{\mathcal{F}}] = \overline{\mathbf{Q}}_u[:, j] \quad j = 1,\dots,M$$

and

$$\overline{\mathbf{r}}^{FE}(\overline{\mathbf{Q}}_u[:, j])[\mathcal{F}^\star, :] = 0 \quad j = 1,\dots,M$$

If $\alpha > 0$ and $\epsilon_{tol} = 0$, then

$$\mathbf{a}^{FE}(t_j,\boldsymbol{\mu})[\overline{\mathcal{F}}] = \overline{\mathbf{V}}\,\mathbf{b}^{FE}(t_j,\boldsymbol{\mu}) \quad j = 1,\dots,M,$$

with

$$\mathbf{b}^{FE}(t_j,\boldsymbol{\mu}) = \overline{\mathbf{V}}^T\,\overline{\mathbf{Q}}_u[:, j] = \overline{\mathbf{V}}^T\,\overline{\mathbf{V}}_u\,\mathbf{b}_u(t_j) \quad j = 1,\dots,M$$

Then,

$$\overline{\mathbf{r}}^{FE}(\overline{\mathbf{V}}^H\,\mathbf{b}^H(t_j,\boldsymbol{\mu}))[\mathcal{F}^\star, :] = 0 \quad j = 1,\dots,M$$

with $\mathbf{b}^H(t_j, \boldsymbol{\mu}) = [(\mathbf{b}^{FE}(t_j, \boldsymbol{\mu}))^T, \mathbf{0}_{\mathcal{R}}^T]^T$. Thus, $\mathbf{b}^H(t_j, \boldsymbol{\mu})$ is the unique solution of the hybrid hyper-reduced equations, and the exact projection of the restrained FE solution. □

The last theorem does not imply that imposing $\mathbf{a}^{FE}(t_j, \boldsymbol{\mu})[\mathcal{I}] = \overline{\mathbf{Q}}_u[\mathcal{I}^*, j]$ as a boundary condition to degrees of freedom in $\mathcal{I}^*$ is the best way to fulfill FE balance equations on the full mesh. In fact, with the additional boundary conditions on $\mathcal{I}^*$, the maximum of available FE equations is card($\mathcal{F}^*$). Theorem 4 means that if the empirical reduced basis is exact, then all the $N_d$ FE balance equations are fulfilled in $\Omega$. In a sense, in the proposed calibration protocol, we better trust in FE balance equations than in experimental data. Accurate FE balance equations can be obtained by a convenient mesh of $\Omega$, although noise is always present in experimental data.

### 5.4. Parameters Updating

In the optimization loop (Figure 8), a given set of parameters $\boldsymbol{\mu}$ is assessed. The H²ROM calculations provide the reduced coordinates associated with the empirical basis previously determined on the RED denoted $\mathbf{b}^H(\boldsymbol{\mu})$. The top reaction $\mathbf{F}^{FE}(\boldsymbol{\mu})$ is also calculated as the average axial stress in the ZOI.

In the example, the cost function that must be minimized, evaluates two scales of error: the microscale error between experimental and computed reduced coordinates and the macroscale error between the measured and computed top reactions. These error functions are, respectively, denoted $\chi_u^2(\boldsymbol{\mu})$ and $\chi_F^2(\boldsymbol{\mu})$.

The microscale error is defined as:

$$\chi_u^2(\boldsymbol{\mu}) = (\mathbf{b}^H(\boldsymbol{\mu}) - (\overline{\mathbf{V}}^H)^T \overline{\mathbf{V}}_u \mathbf{b}_u)^T (\mathbf{b}^H(\boldsymbol{\mu}) - (\overline{\mathbf{V}}^H)^T \overline{\mathbf{V}}_u \mathbf{b}_u) \tag{45}$$

The choice of the norm is user-dependent. The inverse covariance matrix of the displacement is the best norm for a Gaussian noise according to [47,48] for a Bayesian framework. However, in this present study, to keep the treated problem rather simple, a 2-norm has been chosen. The macroscale error is defined as:

$$\chi_F^2(\boldsymbol{\mu}) = \|\mathbf{F}^{FE}(\boldsymbol{\mu}) - \mathbf{F}\|_{\partial_u \Omega}^2 \tag{46}$$

Here, $\partial_u \Omega$ is the top surface of the ZOI, where the experimental load was measured and where the experimental displacements are imposed as Dirichlet boundary conditions. The experimental load measurements are supposed uncorrelated and their variance is denoted by $\sigma_F^2$. In a Bayesian framework, for a Gaussian noise corrupting the load measurements [23], the previous equation can be written as:

$$\chi_F^2(\boldsymbol{\mu}) = \frac{1}{N_t \sigma_F^2} (\mathbf{F}^{FE}(\boldsymbol{\mu}) - \mathbf{F})^T (\mathbf{F}^{FE}(\boldsymbol{\mu}) - \mathbf{F}) \tag{47}$$

For the the optimization loop, the final objective function is a weighted sum of the two previous sub-objective functions:

$$\chi^2(\boldsymbol{\mu}) = c_u \chi_u^2(\boldsymbol{\mu}) + c_F \chi_F^2(\boldsymbol{\mu}) \tag{48}$$

where $c_u$ and $c_F$ are the weights. They can be chosen to balance the two cost functions or to privilege one scale to another. In the illustrating example, the cost function is balanced. A classical Levenberg–Marquardt algorithm is employed for the minimization of the error function and the update of the parameters vector $\boldsymbol{\mu}$.

### 5.5. Model Calibration and FEM Validation

The optimization loop took 53 iterations. The speed ratio between FEM calculations and H²ROM predictions is around 70. Moreover, the H²ROM predictions only needed around 3% of the FEM calculation memory cost. The H²ROM predictions converge way more easily than the FEM calculations. The problem simulated in the optimization loop is a displacement imposed problem. The use of the

reduced basis to predict the displacement field facilitates drastically the convergence. That explains also the important speed-up time that does not come only from the reduction of the integration domain.

Figure 12 displays the experimental and the computed top reactions (initial and optimized). At the end of the optimization loop, it is mandatory to assess the relevance of the H$^2$ROM prediction. The FEMU-H$^2$ROM is dependent on the initial guess $\mu^0$. This input determines the relevance of the reduced basis of the model after the parameters sensibility study and the DEPOD analysis. When updating the model, the parameter set may be too different from the initial guess. As a consequence, the empirical reduced basis $\overline{\mathbf{V}}^H$ may not be accurate and the H$^2$ROM predictions will not be admissible. That is to say that the discrepancy between hyper reduced and Finite Element calculations may not be negligible. That is why the optimized parameters set $\mu^*$ must be validated with FEM calculations on the full domain $\Omega$. It is worth noting that, if the experimental data are included in the DEPOD, the final H$^2$ROM prediction should be close to the experiments.

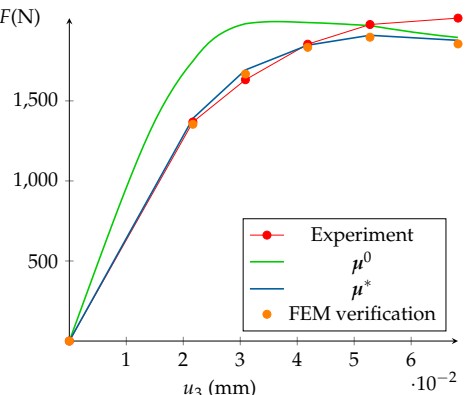

**Figure 12.** Result of the H$^2$ROM optimization.

In a similar manner to the optimization loop, an error function between both calculations can be defined focusing on the microscale (displacement error) and macroscale (top reactions differences).

Concerning the microscale, the discrepancy is only computed in the RED, as H$^2$ROM predictions are only made on this domain and cannot be reconstructed in the full domain with this particular approach. The microscale discrepancy is estimated by $r_u$:

$$r_u^2(\mu^*) = \left(\overline{\mathbf{a}}^H(\mu^*) - \mathbf{a}^{FE}(\mu^*)[\overline{\mathcal{F}}]\right)^T \left(\overline{\mathbf{a}}^H(\mu^*) - \mathbf{a}^{FE}(\mu^*)[\overline{\mathcal{F}}]\right), \text{ with } \overline{\mathbf{a}}^H(\mu^*) = \overline{\mathbf{V}}^H \mathbf{b}^H(\mu^*) \qquad (49)$$

In the same manner, the macroscale discrepancy measure the norm of the difference between the two prediction of the load applied to the specimen. This indicator is denoted by $r_F$. The microscale and macroscale errors should not exceed a few percents of the FEM calculations. In Figure 12, the FEM top reaction is plotted in orange. It is clear that its value is extremely close to the one computed thanks to H$^2$ROM. The error is around 1% at each step.

This final verification is purely numerical. If the H$^2$ROM predictions are validated, it is advised to analyze deeper the full field FEM calculation.

In the case of notable differences between H$^2$ROM prediction and FEM calculations, or between FEM calculations and experiment, the FEMU-H$^2$ROM is not validated. Two solutions are possible to overcome this issue:

1. Perform again the whole parameters sensibility study with $\mu^0 = \mu^*$.
2. Concatenate the previously determined matrix $\overline{\mathbf{X}}$ from Equation (42) with $\overline{\mathbf{Q}}_u(\mu^*)$ and perform a new truncated SVD to determine ultimately an enriched reduced basis $\overline{\mathbf{V}}^H$. No new FEM calculations are needed.

The first solution should be performed in the case of strong differences between H$^2$ROM prediction and FEM calculations. The second option "only" costs a FEM calculation. It is also possible to modify the optimization loop to include regularly FEM-H$^2$ROM comparison and enrich $\overline{\mathbf{V}}^H$ incrementally.

## 6. Discussion

### 6.1. Limitations of the Pruning Procedure

The present paper focused on DVC sets and not on the images themselves. Since each element covers several voxels, the images are also known to be particularly heavy and perhaps more problematic than the DVC data. The pruning procedure considers that they can be deleted. Actually, it can be problematic. For instance, new DVC algorithm could improve the determination of the displacement field (for example for complex problems involving cracks).

The images could be pruned too, in the sense that the only the pixels of the images inside the determined RED can be conserved. However, we preconize to store only the reduced DVC data when the data storage is an issue.

In the case of non homogeneous materials, the data concerning the inhomogeneity outside the RED must be saved as well.

### 6.2. About the Reconstruction of Data outside the RED

Because of the proposed data pruning, experimental data outside the RED are no more available. However, the finite element verification gives access to an estimation of these data via the finite element model and the optimal parameters $\boldsymbol{\mu}^*$. For instance, the shear strain distribution can be estimated by the finite element model with the optimal values of the parameter. In the illustrating example, the computed and measured shear strain distributions, over the integration points in $\Omega$, were compared. The analysis is summarized in the histograms displayed in Figure 13 for the last pre-peak step. The discrepancy between computed (via FE verification) and measured distributions was considered here as satisfying.

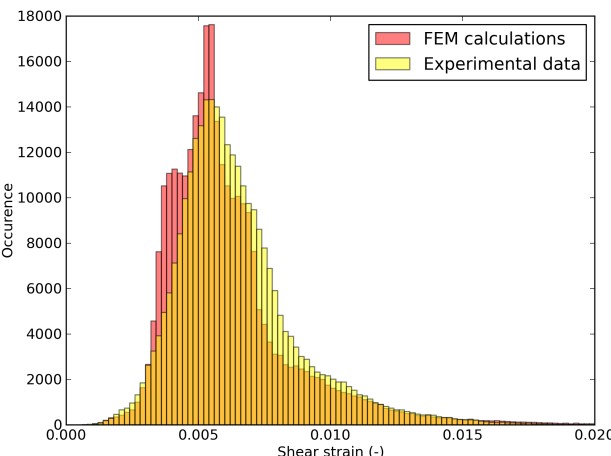

**Figure 13.** Probability distribution of shear strain at the last pre-peak step in the whole domain $\Omega$, comparing FEM calculation (verification step) and experimental data.

### 6.3. Shear Strain Distributions in the RED

We can also consider the shear strain distributions is inside the whole domain $\Omega$ and the RED $\Omega_R$ for the illustrating example. It would be preferable that the pruning procedure stores in the RED the most different configurations. The shear strain distributions in the whole domain and in the RED

might be different (not the same mean value for example). Figures 14a and 15a present the shear strain distributions at the first and last pre-peak step. It appears that the statistical distribution of the shear strain inside the RED is not the same than the one inside the full domain. Nevertheless, zooms at both histograms in Figures 14b and 15b reveal that the extremum values of the shear strain are conserved. One can see that the RED contains nearly all the elements where the shear is maximal. Even if the proposed procedure is model-free, it is intimately linked with the mechanics of solids: it will store preferably the data that are mechanically more relevant. For strain localization phenomenon, it is the most sheared zone. The proposed method is not statistical: it actually induces a sampling bias.

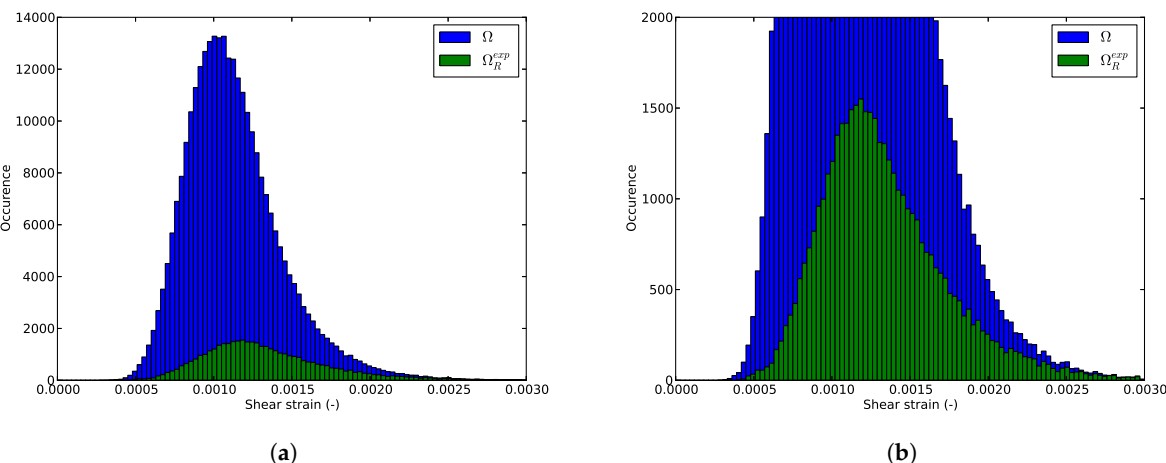

**Figure 14.** Shear strain distributions (**a**) in the whole domain and (**b**) in the RED at the first step.

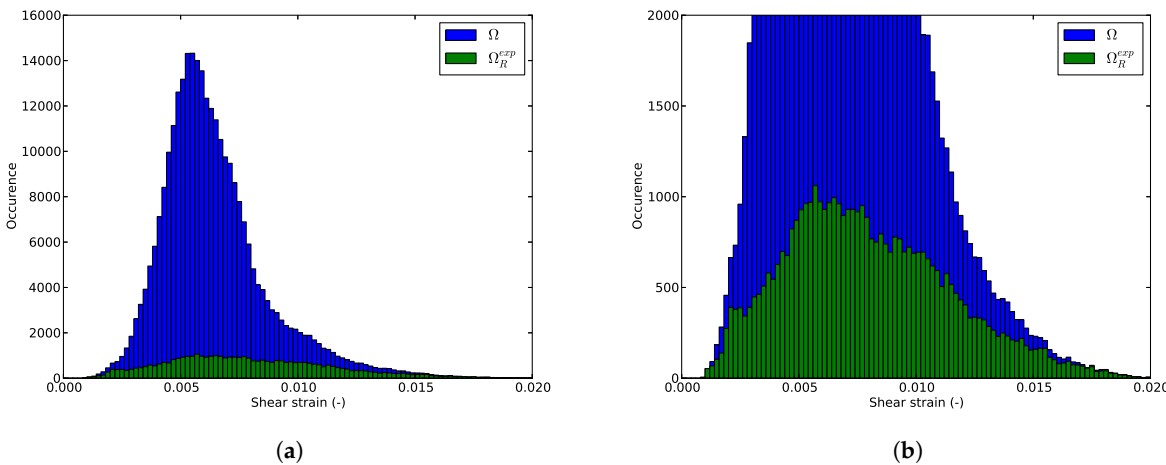

**Figure 15.** Shear strain distributions (**a**) in the whole domain and (**b**) in the RED at the last step.

## 7. Conclusions

The present paper proposes a data pruning procedure for DVC data that is model free and versatile. The k-SWIM algorithm, through its parameter *k*, enables the user to define the size of the stored data.

The resultant data can still be used afterwards, for instance for calibration. The use of hybrid hyper-reduction is particularly suitable for the pruned data as it enables a non-negligible reduction of memory and time costs in the FEMU optimization loop. The FEMU-H$^2$ROM method is thus a new way to use massive DVC data for deeper mechanical studies.

**Supplementary Materials:** The following are available online at http://www.mdpi.com/2297-8747/24/1/18/s1 as supplementary files to make the output of Algorithm 1 reproducible. The ASCII file Node-iXYZ.txt contains the node indices and the related coordinates. The files Vu.npy, bu.npy and Pu_reference.npy, are binary files related to $\mathbf{V}_u$, $\mathbf{b}_u$ and $\mathcal{P}_u$, respectively. They have been generated by using the NumPy instruction "save". The ASCII file k_swim.py contains Algorithm 1 written with SciPy instructions. In the ASCII file run_kswim.py, this algorithm is applied to the data $\mathbf{V}_u$.

**Author Contributions:** Conceptualization, D.R.; methodology, D.R. and W.H.; experimental data, C.M.; data analysis, W.H.; writing, D.R. and W.H.

**Funding:** This research was funded by Agence Nationale de la Recherche, in France; grant number is ANR-14-CE07-0038-03 FIMALIPO.

**Acknowledgments:** The authors would like to acknowledge the Agence Nationale de la Recherche for their financial support for the FIMALIPO project.

**Conflicts of Interest:** The authors declare no conflict of interest.

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
