# Peer review of "Data Pruning of Tomographic Data for the Calibration of Strain Localization Models"

_mca, doi:10.3390/mca24010018_

Round 1
Reviewer 1 Report
Please see the pdf file.

Author Response
The pruning algorithm is better explained in the introduction of the paper by adding a flowchart. As explained now in this figure, in the introduction and in the abstract of the paper, the proposed data pruning concerns experimental data only. The main contributions of the paper are shown in Figure 1, inside a dashed rectangle. Now, the algorithm shown in the result part (Figure 8) is focused on the calibration procedure, not on the pruning itself. As explained in the result section, the calibration is done via the hybrid hyper-reduction with a reduced domain imposed by the pruning of the experimental data. Contrary to the usual hyper-reduction method, this reduced domain is not obtained by using simulation data, as explained in the introduction page 3 line 94 and in Section 2.3 page 6 line 149. We agree that the FEMU-H2ROM is an application of the hyper-reduction method with a standard finite element updating procedure, except the construction of the reduced integration domain. This calibration procedure aims to show that pruned data allow model calibration although only few experimental data are saved in a storage system (see in the introduction page 3 lines 86 to 90).
In order to make results on pruning reproducible, the following data are now available as "Supplementary Files" :
The node positions, Vuexp, buexp and Pu. Details on the content of the file are given in Appendix A.
We believe that the illustration is relevant because data pruning make sense only on noisy data with gradient that have to be explained by a model to calibrate. Here the constitutive equations enable to replicate the gradient of displacements observed in the experimental data. In addition, model calibration via hyper-reduction makes sense only for nonlinear constitutive equations. Now, all the data required for the model calibration are available, assuming that the proposed constitutive equations are implemented in a finite element solver. So these data are reusable.
The pruning does not show “useful results for physics” because its purpose is to free memory in storage systems used in material science, wile preserving the capabilities for model calibration (see the introduction lines 22-29).
Concerning the novelty of the algorithm itself, the main difference with other model order reduction methods is that the reduced domain used for the projection of the equations, is not generated by using simulation data (see the introduction line 94-95). It is imposed by the pruning procedure applied to experimental data prior any simulation. It should be clearer now with Figure 1. This is a major difference with the POD-DEIM and the method proposed in Rowley2005 since these methods are using simulation data only. To our knowledge, the POD-DEIM method aims set-up reduced equations on interpolation points selected by using simulation results. Then comparison would not make sense here.
Because nonlinear behavior are assumed to be involved in the experimental data the question of finding a convenient metric that explain the relevance of the data pruning is a very hard task. Our answer is, if the model calibration is still feasible after pruning, than the data pruning is relevant (see the beginning of Section 2.3). We acknowledge that this answer is not fully satisfactory and deserve more research.
1. General comments
Two references [1] and [2] have been added in the introduction so that the reader can read about what image or volume correlations are.
We have explained the following terms:
- Constitutive model: it is a set of closure equations in solid mechanics
- Data pruning has been defined in the introduction
- Non-isotropic materials have mechanical properties that depend on their orientation.
- Reduced integration domain: it is a subdomain of a body, where the reduced equations are set up.
The term optical flow has been removed from the paper.
2. Introduction
The goal is better explained in the introduction in lines 22 to 29, lines 86 to 90 and at the beginning of Section 2.
Acronyms have ben corrected.
Few words have been added about the lack of reconstruction after pruning, at line 78 and in Section 5.2.
3. Data pruning
The goal of the method is now explained with few introductory words in Section 2.
To reduce the length of this section we have removed the definition of the local error h.
The term “objective function” has been removed.
The letter K has been replaced by a lowercase letter k, in the description of the k-SWIM algorithm.
4. Illustrative example
The location of the selected zones depends on the experimental data. We don’t have any explanation why these zones are selected.
In the paper, the final RED was arbitrary selected with k=25, as explained in Section 3.4.
We did not found a convenient metric that explain the relevance of the data pruning. The pruned data are assumed to be relevant if the model calibration is still feasible after pruning (see the beginning of Section 2.3 ).
A better toy system must allows:
- parameters calibration for constitutive equations (closure equations)
- experimental data showing local gradients that are not solely noise acquisition
- nonlinear finite element equations such that hyper-reduction make sense during the calibration procedure and such that strain localization can be replicated by the numerical simulations
- X-ray tomography
We think that resin bounded sands are convenient for that purpose, with the given experimental data as supplementary files.
5. FEMU-H2ROM
1 - The flowchart in Figure 8 is better explained now at lines 305 to 309.
2- The assertions related to the mechanical properties of sand cores where not developed in this paper because it is not the main focus of the paper.
3- Simulation data and experimental data are fused during the FEMU-H2ROM, in the matrix \overline{X} (Equation (42)).
4- The FEMU-H2ROM has been introduced to make the calibration procedure consistent with the lack of experimental data due to the data-pruning, as explained in the introduction lines 28, 78, 93 and at line 116 in Section 2.
5,6- We did not performed a deep statistical analysis of the strain distribution in the specimen neither in the FE results computed for the verification procedure. This is not the focus of the paper.
7- As explained in the introduction part, the gain of data pruning is to free storage memory in order to save other experimental data related to new experiments. Compared to other pruning method, we propose a model free approach.
8- The title of sections 4 and 4.1 were not appropriate. This has been changed now.
The pruning procedure is now introduced in Section 1.
6. References
The 2 proposed references have been added to the paper, and other references have been corrected.

Reviewer 2 Report
see enclosed document

Author Response
We have simplified the section 2.3 and 2.4.
Reply to the remarks:
1. the proposed sentence has been add as a comment in Algorithm 1. We acknowledge that the k-SWIM algorithm may provide non unique output. Especially when considering the M first vectors of a natural basis as input. For l>1, the algorithm will select randomly the indices after the first selected index. To enforce the extension of \mathcal{P}_{j-1}, we have modified the algorithm in order to exclude this set of indices when searching for a new maximum. The following comment have been added at the beginning of Section 2.3, in order to explain why we are using an extension of the DEIM algorithm to create the RED: We are not able to prove that the proposed approach has a strong physical basis for pruning data according to an appropriate metric. The proposed approach is heuristic, but it fulfills some mathematical properties.
The reason why we do not consider interpolation is clearly explained now in Section 2.3 line 172.
In the proof, line 181, the 2-norm of the residual is higher than maximum of the absolute value of this vector. So the proof is correct.
2. For the sake of simplicity we have removed considerations about h.
3. The reduced mesh is the mesh of the RED. This has been clarified in the revised paper, above Equation (24). As recommended by the reviewer, a new figure (Figure 2.) has been added to show a schematic view of the various sets of indices. A comment has been added to the assumption in equation (26). It has been extended to nonlinear cases too. The assumption in theorem 4 is related to the ability to reconstruct the FE solution by using the data in X and the FE solution restricted to the RED. Theorem 4 has been clarified on that point. Sets of indices related to the reduced mesh are now shown in the Figure 2.
4. We have rewritten the proof of theorem 5.
Minor corrections
1. We have changed \ref by \eref when referring to an equation.
2. Correction done
3. Correction done
4. Correction done
5. Correction done.
6. Correction done.
7. Correction done. In addition we have replace q by r.
8. \overline{N} has been replace by NR
9. Correction done
10. Correction done
11. \mathcal{H} has been replaced by \mathcal{R}.
12. This subdomain is W+ defined in eq. (16).
13. Correction done
14. This is the Frobenius norm. Correction done.
15. Correction done
16. This is the magnitude of the displacement.
17. Correction done
18. The proposed sentence is: “When updating the model, the parameter set may be too different from the initial guess.”
19. The definitions have been included in a sentence.
20. The top of the specimen, where the load is measured, is now denoted $\partial_F \Omega$.
21. The proposed sentence is: Since each element covers several voxels, the images are known to be as-well particularly heavy and perhaps more problematic than the DVC data.
22. Correction done
